# Mass Production of Rg1-Loaded Small Extracellular Vesicles Using a 3D Bioreactor System for Enhanced Cardioprotective Efficacy of Doxorubicin-Induced Cardiotoxicity

**DOI:** 10.3390/pharmaceutics16050593

**Published:** 2024-04-26

**Authors:** Yunfeng Di, Shuang Zhao, Huilan Fan, Wei Li, Guangjian Jiang, Yong Wang, Chun Li, Wei Wang, Jingyu Wang

**Affiliations:** 1College of Traditional Chinese Medicine, Beijing University of Chinese Medicine, Beijing 100029, Chinabucmjiang@bucm.edu.cn (G.J.);; 2College of Traditional Chinese Medicine, Xinjiang Medical University, Urumqi 830017, China; 3Key Laboratory of TCM Syndrome and Formula, Beijing University of Chinese Medicine, Ministry of Education, Beijing 100029, China; 010700@gzucm.edu.cn; 4Modern Research Center for Traditional Chinese Medicine, Beijing University of Chinese Medicine, Beijing 100029, China; 5State Key Laboratory of Traditional Chinese Medicine Syndrome, Guangzhou University of Chinese Medicine, Guangzhou 510006, China

**Keywords:** mesenchymal stromal cells, 3D cell culture, small extracellular vesicles, drug delivery, Ginsenoside Rg1, cardioprotective efficacy

## Abstract

Background: Small extracellular vesicles (sEVs) obtained from human umbilical cord mesenchymal stromal cells (MSCs) have shown cardioprotective efficacy in doxorubicin-induced cardiotoxicity (DIC). However, their clinical application is limited due to the low yield and high consumption. This study aims to achieve large-scale production of sEVs using a three-dimensional (3D) bioreactor system. In addition, sEVs were developed to deliver Ginsenoside Rg1 (Rg1), a compound derived from traditional Chinese medicine, *Ginseng*, that has cardioprotective properties but limited bioavailability, to enhance the treatment of DIC. Methods: The 3D bioreactor system with spinner flasks was used to expand human umbilical cord MSCs and collect MSC-conditioned medium. Subsequently, sEVs were isolated from the conditioned medium using differential ultra-centrifugation (dUC). The sEVs were loaded with Ginsenoside Rg1 by electroporation and evaluated for cardioprotective efficacy using Cell Counting Kit-8 (CCK-8) analysis, Annexin V/PI staining and live cell count of H9c2 cells under DIC. Results: Using the 3D bioreactor system with spinner flasks, the expansion of MSCs reached ~600 million, and the production of sEVs was up to 2.2 × 10^12^ particles in five days with significantly reduced bench work compared to traditional 2D flasks. With the optimized protocol, the Ginsenoside Rg1 loading efficiency of sEVs by electroporation was ~21%, higher than sonication or co-incubation. Moreover, Rg1-loaded sEVs had attenuated DOX-induced cardiotoxicity with reduced apoptosis compared to free Ginsenoside Rg1 or sEVs. Conclusions: The 3D culture system scaled up the production of sEVs, which facilitated the Rg1 delivery and attenuated cardiomyocyte apoptosis, suggesting a potential treatment of DOX-induced cardiotoxicity.

## 1. Introduction

Doxorubicin (DOX) is commonly used in chemotherapy for breast cancer and lymphoma, but it is associated with cardiotoxicity [1]. Currently, the only FDA-approved drug for doxorubicin-induced cardiotoxicity (DIC) is dexrazoxane [2]. However, its use in clinical practice is limited due to its side effects and strict indications [3]. Therefore, there is an urgent need to develop more effective and safer treatments for DIC.

The small extracellular vesicles (sEVs) derived from mesenchymal stromal cells (MSCs) have been widely used in cardiovascular treatments to protect damaged cardiac cells from apoptosis [4]. The sEVs are double-layered phospholipid membrane structures less than 200 nm in diameter that are biocompatible, low immunogenic and capable of crossing biological barriers, making them ideal for delivery of various types of drugs, including nucleic acids, proteins and small molecules [5,6,7]. However, the low yield and high consumption of sEVs have hindered their applications. The dose of sEVs for the treatment of cardiac injury in mice or rats ranges from 200 μg to 2 mg/kg, which requires over one million cells for sEV collection per mouse, and around a billion cells in potential clinical applications [8,9]. However, the traditional two-dimensional (2D) culture takes lots of bench work and is hardly capable of large-scale production of MSC-derived sEVs. Therefore, there is an urgent need to develop a mass production system and to enhance the treatment efficiency of MSC-sEVs for clinical applications.

Nowadays, three-dimensional (3D) configurations have been applied in automated large-scale cell cultures. Polysaccharide carriers are commonly used in bioreactors to collect cellular secreted products due to their ability to support cell adhesion and growth on the surface [10]. To increase the efficiency of cell expansion, novel porous protein microcarriers have been developed, which have an increased surface area to volume ratio due to their internally interconnected porous structure, facilitating cell growth both on the surface and inside, resulting in a significant increase in cell growth area, allowing for rapid harvesting of cells and sEVs [11]. 

Herein, sEVs were applied to load and deliver Ginsenoside Rg1 (Rg1) for improved therapeutic efficacy of DIC [12]. Ginsenoside Rg1 is a compound isolated from Renshen, also known as *Ginseng Radix et Rhizoma* in traditional Chinese medicine [13]. Rg1 has been shown to have a protective effect on cardiomyocytes after various types of damage, including damage induced by inflammatory factors and oxidative stress [14]. However, its clinical application is limited due to low oral bioavailability and short half-life [15,16]. Therefore, a novel Rg1-based therapy by sEV encapsulation has great potential to reduce cardiomyocyte apoptosis under DIC. 

This study utilized a bioreactor system with porous protein carriers to scale up the production of MSCs and sEVs with reduced bench work, as compared to traditional 2D culture. Additionally, sEVs were loaded and optimized to deliver Rg1, which protects damaged cardiomyocytes from apoptosis with improved efficiency, for the development of potential novel treatments of DIC.

## 2. Materials and Methods

### 2.1. Cell Culture

Human umbilical cord MSCs (passage 2) were purchased from CytoNiche Biotechnology. Cells were incubated and expanded in T75 cm^2^ flasks (Corning, Corning, NY, USA) for monolayer culture with the serum-free MSC culture medium (CytoNiche, Beijing, China). The serum-free MSC culture medium contains no ingredients of animal origin and does contain platelet lysate. Cells grew to 80~90% confluence and were harvested with 0.05% Trypsin-EDTA (Gibco, Grand Island, NY, USA) for cell subculture and expansion (8000 cells/cm^2^) at 37 °C in a humidified atmosphere with 5% CO_2_. Passage 4 cells were loaded to microcarriers for 3D rotating culture.

### 2.2. SEM Imaging of Microcarriers

The morphology of 3D microcarriers in tablets (CytoNiche, Beijing, China) was imaged using scanning electron microscopy (SEM, Thermo Fisher Scientific, Waltham, MA, USA). The tablets were dispersed in deionized water, frozen at −20 °C for 2 h, and lyophilized for 5–6 h to obtain dry microcarriers in powder form for imaging.

### 2.3. Scale-up Expansion of MSCs with Bioreactor System

Firstly, 2.15 g of porous microcarriers were soaked in MSC medium in the 500 mL spinner flask overnight at 4 °C. Secondly, the soaked microcarriers and 2 × 10^7^ MSCs were added to the spinner flask on the following day, which is referred to as Day 0. To improve cell seeding in the microcarriers, the monitor was set to a cycle of 40 rpm for 5 min and 1 rpm for 2 h, which lasted for 24 h. On Day 1, samples from four flasks were prepared and the microcarriers were dissociated with lysis solution to determine cell number and viability using trypan blue staining. These steps were repeated on Days 3, 4 and 5. Finally, on Day 5, the MSC-conditioned medium was harvested for isolation of 3D-sEVs. At the same time, MSCs were cultured in 2D T75 flasks and 2D-MSC conditioned medium was collected for 2D-sEV isolation. The conditioned medium was stored at −80 °C before sEV isolation.

### 2.4. Trypan Blue Staining

Cell viability in the 2D and 3D cell culture systems was determined by trypan blue staining. The 2D-MSCs and 3D-MSCs were suspended and stained with a trypan blue solution (Solarbio, Beijing, China). Then, the number and viability of living cells were counted by the cell counter (Countstar, Shanghai, China).

### 2.5. Isolation of sEVs

The MSC-conditioned medium was centrifuged at 300× *g* for 10 min at room temperature (RT) to remove cells, then 2000× *g* for 10 min at 4 °C to remove cellular debris. After that, the supernatant was centrifuged at 10,000× *g* for 30 min at 4 °C to remove microvesicles. The sEVs were isolated by ultracentrifuging twice at 135,000× *g* for 90 min (Beckman, Brea, CA, USA). The purified sEVs were harvested in 300 μL of phosphate-buffered saline (PBS) (HyClone, Logan, UT, USA) and stored at −80 °C. 

### 2.6. Characterization of sEVs

After sEV isolation, the morphological structure of sEVs was observed by transmission electron microscopy (TEM, Hitachi, Tokyo, Japan). After 2% uranyl acetate negative staining, we measured the size, distribution and concentration of sEVs with nanoparticle tracking analysis (NTA, ZetaView, Munich, Germany). The protein concentration of sEVs was quantified by a BCA protein assay kit (Applygen, Beijing, China) according to the manufacturer’s instructions. The expression of markers was detected using Western blot (WB) analysis with anti-CD63, anti-CD81 and anti-TSG101 antibodies (Abcam, Cambridge, UK) [17]. 

### 2.7. DOX Induced-Myocardial Injury Model 

H9c2 cells, a cardiomyocyte cell line, were obtained from China Infrastructure of Cell Line Resources. H9c2 cells were cultured in Dulbecco’s modified Eagle’s medium (Gibco, Grand Island, NY, USA) supplemented with 10% fetal bovine serum (Gibco, Grand Island, NY, USA) and 1% penicillin-streptomycin (Gibco, Grand Island, NY, USA) at 37 °C with 5% CO_2_ for monolayer confluency of about 80%~90%. Cells were harvested with 0.25% Trypsin-EDTA (Gibco, Grand Island, NY, USA) and seeded (8000 cells/well) into 96-well plates for 24 h, then treated with DOX (Yuanye, Shanghai, China) for 24 h at desired concentrations.

### 2.8. Drug Loading of sEVs

To establish the drug delivery system of sEVs, three methods were used to encapsulate Ginsenoside Rg1 (Yuanye, Shanghai, China) into sEVs, including electroporation, sonication and co-incubation.

In the electroporation method, sEVs and Rg1 were mixed in a 15:1 volume ratio to form an sEV-Rg1 mixture. Then, the mixture was added to a 4 mm electric shock cup. The sEV-Rg1 mixture was electroporated using a pulse width of 0.1 ms and 10 pulses at 560 V for 5 cycles. The mixture was then co-incubated at 4 °C for 24 h.

In the sonication method, sEVs and Rg1 were mixed in a 15:1 volume ratio to form the sEV-Rg1 mixture, and it was added to the ultrasound tube. The ultrasound tube was applied 5 cycles of 30 s on/off for 3 min with a 2 min cooling period between each cycle. The mixture was then co-incubated at 37 °C for 1 h, then stored at 4 °C.

In the co-incubation method, sEVs and Rg1 were mixed in a 15:1 volume to form sEV-Rg1 mixture, and then were co-incubated at 4 °C for 48 h. 

### 2.9. Drug Loading Efficiency of sEVs

To evaluate the drug loading efficiency of the three methods, the weight of total Rg1 and Rg1 in the supernatant after UC at 135,000× *g* for 90 min were detected by high performance liquid chromatography (HPLC, Thermo Fisher Scientific, Waltham, MA, USA). 

Before HPLC detection, the total Rg1 group mixture was separately mixed with methanol in equal proportions, ultrasonicated for 30 min and centrifuged at 15,000× *g* rpm for 10 min to remove the lipid bilayer of the sEVs. 

The drug loading efficiency was calculated by the following equation:$$M_{Rg1}=\frac{A_{Rg1}\times C_{s}\;}{A_{s}\;}\times V_{Rg1}\;$$
$$Drug\;loading\;efficiency\left(\%\right)=\frac{M_{t}-M_{f}\;}{M_{t}}\times 100\%$$
*M_Rg_*_1_: the total weight of the Rg1 sample. *A_Rg_*_1_: the peak area of Rg1 sample. *A_s_*: the peak area of Ginsenoside Rg1 standard. *C_s_*: the concentration of Ginsenoside Rg1 standard. *V_Rg_*_1_: the volume of Rg1 sample. *M_t_*: the weight of total Rg1. *M_f_*: the weight of free Rg1 in the supernatant.

The oral bioavailability data of Ginsenoside Rg1 were obtained from the website Lab of Systems Pharmacology. 

### 2.10. Cell Viability Assay Using CCK-8

Cell viability was detected using the CCK-8 kit according to the manufacturer’s instructions (Vazyme, Nanjing, China). After treatment of H9c2 cells with DOX for 24 h, cells were mixed with the CCK-8 reagent and maintained for 2 h at 37 °C in 5% CO_2_. The absorbance under 450 nm was detected using a microplate reader (Molecular Devices, Sunnyvale, CA, USA).

### 2.11. Flow Cytometry Assay

To characterize the expression of surface markers, the MSCs were immune-stained and detected by flow cytometry analysis. Antibodies used in this detection are anti-CD73-PE, anti-CD90-FITC, anti-CD105-PE, anti-CD14-FITC, anti-CD34-FITC, anti-CD19-PE, anti-CD45-APC and anti-HLA-DR-PE (Biolegend, San Diego, CA, USA). 

To assess the apoptotic rates, H9c2 cells were seeded in a 6-well plate at 30,000 cells/well for 24 h and then desired treatments were performed for 24 h. Cells were stained using an Annexin V/PI staining kit (Beyotime, Shanghai, China) following the manufacturer’s instructions and detected by flow cytometry (BD Biosciences, Franklin Lakes, NJ, USA).

### 2.12. Live Cell Imaging and Analysis

H9c2 cells were seeded in 96-well plates at 8000 cells/well, and imaged using ImageXpress Pico automated cell imaging system (Molecular Devices, Sunnyvale, CA, USA) for 24 h of continuous fluorescence or brightfield imaging. Cells were then segmented and counted using the CellReporterXpress system. 

### 2.13. sEV Uptake Essay

The sEVs were labelled with Dil dye (Beyotime, Shanghai, China) after 15 min incubation at 4 °C. The stained sEVs were co-incubated with H9c2 cells for 6 h at 37 °C. The H9c2 cells were then labelled with Calcein-AM dye (Beyotime, China) for 10 min at 37 °C and imaged by confocal laser scanning microscopy (Leica, Wetzlar, Germany).

### 2.14. Statistical Analysis and Drawing

All data were presented as the mean ± standard deviation. Statistical analysis between multiple groups was performed by one-way ANOVA or two-way ANOVA using GraphPad Prism 6 (San Diego, CA, USA). Differences were considered statistically significant at *p* < 0.05.

The schematic figures in the Figure 6A were completed using Figdraw 2.0 URL: https://www.figdraw.com/static/index.html#/, accessed on 9 April 2024.

## 3. Results

### 3.1. Three-Dimensional Rotating Bioreactor Scaled-up MSC Culture

The sEVs can be used as vehicles for drug delivery and have significant cardioprotective effects. Nevertheless, their low yield and high consumption have limited their application and research. To achieve large-scale production of sEVs for the treatment of DOX-induced cardiotoxicity, the 3D cell culture system was selected to culture MSCs derived from the human umbilical cord for the following experimental research (Figure 1A). The 3D cell culture system included porous microcarriers, spinner flasks and a monitor (Figure 1B). The morphology of porous microcarriers was imaged by SEM and the diameter of the microcarrier is about 350 μm. Four 500 mL spinner flasks were used to culture MSCs in the 3D culture system (3D-MSCs); at the same time, five T75 flasks were used in the 2D culture system (2D-MSCs). The 3D-MSCs showed proliferation as indicated by Calcein-AM/PI staining on Days 1, 3, 4 and 5 (Figure 1C). Cell viability increased to over 90% after Day 1 (Figure 1D). The total number of living cells on Day 5 was 600 million in 3D bioreactors, an increase of approximately 7.5-fold compared to Day 0 (Figure 1E). Taken together, the 3D cell culture system achieved large-scale production of MSCs. Meanwhile, the cell number of 2D-MSCs was able to obtain around 8.08 × 10^6^ on Day 3.

To characterize the expression of surface biomarkers, 2D-MSCs and 3D-MSCs were harvested and analyzed using immuno-staining and flow cytometry analysis. One of the standards of MSCs for clinical therapy is that more than 95% of the MSC population must express CD90, CD73 and CD105, while less than 2% of cells expressing CD14, CD34, CD45 or CD11b, CD79a or CD19 and HLA-DR [18]. Similar to that of 2D-MSCs, over 95% of 3D-MSCs expressed positive biomarkers CD73+, CD90+ and CD105+. And no more than 2% of 3D-MSCs expressed negative biomarkers CD14−, CD34−, CD19−, CD45− and HLA-DR− (Figure 2A,B). This validated that the harvested cells from the 3D culture system could maintain MSC features in scale-up expansion.

### 3.2. Three-Dimensional Rotating Bioreactor Enabled Mass Production of sEVs 

The 2D-MSC conditioned medium was collected on Day 3 and 3D-MSC conditioned medium was collected on Day 5. Here, we utilized dUC, the most frequently used method, for 2D-sEV and 3D-sEV isolation (Figure 3A), then we used TEM, WB and NTA for sEV characterization [19]. Both 2D-sEVs and 3D-sEVs showed a morphology with a spherical or oval-shaped appearance, which was consistent with previous reports describing the typical morphologies of sEVs (Figure 3B) [20]. WB analysis revealed that 2D-sEVs and 3D-sEVs expressed CD63, CD81 and TSG101, the markers of sEVs (Figure 3C). The NTA results showed that the peak diameter of 2D-sEVs was 140.8 nm and that of 3D-sEVs was 143.9 nm. The particle concentration in 3D-sEVs was significantly higher than that in 2D-sEVs, suggesting that 3D culture could obtain more particles of sEVs (Figure 3D). These results confirmed that particles harvested from the 3D cell culture system could maintain the character of sEVs like the traditional 2D cell culture system.

In addition, when considering the bench work, with the same number of cells as input (8 × 10^7^ cells), four spinner flasks were needed in 3D-MSC culture, while 132 T75 flasks were required in 2D-MSC culture as calculated based on our data obtained from five flasks (Table 1). The total protein or total particle of 3D-sEVs increased by around 2-fold and the experimental time decreased by approximately 8-fold compared to 2D culture. Meanwhile, the concentration of particles in the MSC complete medium without cells and MSC conditioned medium after culture was detected by NTA. The results showed the former medium contained approximately 1.2 × 10^10^/mL particles, while the latter contained approximately 2.3 × 10^10^/mL particles (Appendix A). It hinted that there was some consumption of the original vesicles in the medium and significant production of new vesicles after MSC culture. These also demonstrated the possibility of the 3D culture system, which can achieve large-scale production of MSCs and sEVs, significantly improving experimental efficiency.

### 3.3. The 3D-sEVs Decreased H9c2 Cell Apoptosis and Protected H9c2 Cells from DOX-Induced Cardio Injury

DOX is a widely used chemotherapy drug in the treatment of breast cancer, lymphoma and others [1]. However, it exhibited cardiovascular cytotoxicity and induced cardiomyocyte apoptosis or ferroptosis [21]. In this study, the therapeutic function of sEVs was verified by assessing H9c2 viability under DOX-induced injury (Figure 4A). The cell viability of H9c2 cells with gradient concentrations of DOX ranging from 1 to 8 μM in H9c2 cells was evaluated using the CCK-8 assay. The results showed that DOX significantly reduced H9c2 cell viability at concentrations between 1 and 8 μM. Because 1 μM DOX reduced H9c2 cell viability to 62%, it was selected for further experiments (Figure 4B). To assess the cellular uptake of sEVs by H9c2 cells, Dil dye was used to label sEVs. After 6 h, Dil-labelled sEVs (red spots) could be observed in the cytoplasm of living H9c2 cells, indicating the quick cellular uptake of sEVs (Figure 4C). The sEVs at 20 μg/mL (equal to 2.5 × 10^10^/mL particles) and 50 μg/mL (equal to 6.25 × 10^10^/mL particles) exhibited significant treatment efficacy to DOX-induced cardio injury. The cell viability of H9c2 cells increased to ~80% with sEV treatment (Figure 4D). The cell apoptosis detected by flow cytometry showed that after sEV treatment, the percentage of Q1 (necrosis) and Q2 + Q4 (late and early apoptosis) decreased, while Q3 (living cell population) increased remarkably, compared to DOX model group (Figure 4E,F). Meanwhile, the apoptosis rate of H9c2 after sEV treatment at 20 μg/mL and 50 μg/mL decreased to ~10% (Figure 4G). Subsequently, decreased apoptosis was visualized using Annexin V/PI staining in the sEV-treated group compared to the DOX model group (Figure 4H). Taken together, 3D-sEVs ameliorated DOX-induced apoptosis in cardiomyocytes and exhibited higher protective efficiency.

### 3.4. Ginsenoside Rg1 Decreased H9c2 Cell Apoptosis and Protected H9c2 Cells from DOX-Induced Cardio Injury

Next, we evaluated the cardioprotective role of Ginsenoside Rg1 in DOX-induced cardio injury. Gradient concentrations of Ginsenoside Rg1 were applied to decrease DOX-induced H9c2 cell apoptosis. The cell viability of H9c2 cells increased by ~10% with treatment with Ginsenoside Rg1 at 5~20 μg/mL compared to the DOX model group. However, there was no significant cardioprotective efficiency of Ginsenoside Rg1 at 25~40 μg/mL (Figure 5A). Based on Annexin V/PI staining, Ginsenoside Rg1 at 5~20 μg/mL significantly increased the percentage of living cells and decreased apoptosis rate compared to the DOX model group (Figure 5B–D). Furthermore, ImageXpress Pico automated cell imaging system was used to track the cell number of H9c2 cells treated with 5~20 μg/mL Ginsenoside Rg1 within 24 h (Figure 5E and Appendix A). The relative cell number after Ginsenoside Rg1 treatment was higher than the DOX model group (Figure 5F). After 24 h, the cell number increased to ~75% after 20 μg/mL Ginsenoside Rg1 treatment (Figure 5G). The results indicated that Ginsenoside Rg1 mitigated cell apoptosis, promoted cell proliferation and protected cardiomyocytes from DOX-induced injury. 

### 3.5. Rg1-Loaded sEVs Reduced DOX-Induced H9c2 Cell Damage

To optimize the sEV-based delivery system of Rg1, the loading efficiency of three methods, i.e., electroporation, sonication and co-incubation, was evaluated using HPLC (Figure 6A). The peak of Ginsenoside Rg1 was observed after 10 min in different groups (Figure 6B). The results exhibited that drug loading efficiency with electroporation was 21% which was higher than sonication (16%) and co-incubation (2%) (Figure 6C). The morphology of Rg1-loaded sEVs maintained the spherical appearance and diameter distribution (Figure 6D,E). Therefore, electroporation was chosen for further research. 

To evaluate the cardio-protection of Rg1-loaded sEVs, the cell viability, apoptosis and proliferation of H9c2 cells were detected. The cell viability of H9c2 treated with sEVs, Rg1 and Rg1-loaded sEVs was also assessed. The results showed that Rg1-loaded sEVs at 10 μg/mL increased by ~15% and that at 20 μg/mL increased by ~40% compared to the DOX group. The cell viability of the Rg1-loaded sEV group was higher than that of sEVs alone or Rg1 alone (Figure 7A and Appendix A). The concentration of Rg1-loaded sEVs was reduced to 10 μg/mL for further experiments. To assess cell number and proliferation of H9c2 cells, the results showed that the relative cell number of H9c2 after Rg1-loaded sEVs and Rg1 treatment was higher than that of the DOX model group based on ImageXpress Pico automated cell imaging system within 24 h (Figure 7B). After 24 h, 10 μg/mL Rg1-loaded sEVs showed comparable cardio-protection with 20 μg/mL free Rg1 (Figure 7C). Similarly, Rg1-loaded sEVs at 10 μg/mL significantly increased the percentage of living cells and decreased apoptosis rate compared to the sEVs and Rg1 group according to the Annexin V/PI staining (Figure 7D,E). The apoptosis rate of Rg1-loaded sEVs at 10 μg/mL decreased to ~9% (Figure 7F). Taken together, Rg1-loaded sEVs showed improved cardioprotective functions compared to free Rg1 and sEVs, suggesting a promising clinical application in future.

## 4. Discussion 

The sEVs derived from MSCs have shown great therapeutic value in cardiovascular, neurodegenerative and other fields [22]. They also serve as natural delivery systems for nucleic acids, proteins and compounds, promoting stability and barrier crossing [23]. However, the clinical application of sEVs faces multiple challenges, and one of the most critical is low yield and high consumption [24]. Our study proposed the use of a 3D rotating bioreactor as a scalable and cost-effective system for the large-scale production of MSC-derived sEVs as carriers of compounds. The spinner flasks’ ease of handling and low time cost would facilitate the smooth transition from lab scale to industrial scale production of sEVs as drug delivery vehicles. 

The compound, Ginsenoside Rg1, derived from traditional Chinese medicine, *Ginseng Radix et Rhizoma*, has shown effective cardio-protection against DOX-induced cardio injuries [25]. However, its oral bioavailability is limited to 9.03% and hinders its further application [15,16,26]. Herein, Rg1 was loaded into MSC-derived sEVs by electroporation. In comparison with free Rg1 and sEVs, Rg1-loaded sEVs improved the cell viability and proliferation of H9c2 cells after DOX treatment. Additionally, Rg1-loaded sEVs significantly decreased DOX-induced cell apoptosis. Based on the drug delivery function, sEVs could deliver DOX compound to increase the antitumor efficacy, the research has shown that DOX-sEVs avoided heart toxicity by partially limiting the crossing of DOX through the myocardial endothelial cells in the treatment of breast and ovarian cancer [27], suggesting potential clinical applications. 

The production of sEVs in this study was performed in a xeno-free medium containing human platelet lysate, which facilitated cell proliferation. However, the sEVs we collected in MSC-conditioned medium could have originated from both MSCs and human platelet lysate [28,29]. It was still a challenge to track and purify MSC-secreted sEVs in production. In addition, further characterization of sEVs is expected, such as quantification of total lipids, total RNA, protein composition, non-protein markers of sEVs and localization of sEV-associated components. Further research is expected to investigate the molecular mechanism and pharmacological kinetics of the enhanced cardioprotection of Rg1-loaded sEVs.

## 5. Conclusions

In summary, our study has successfully demonstrated that a 3D cell culture system based on rotating bioreactors can produce a high yield of MSC-derived sEVs. These 3D-sEVs can be used as carriers to deliver Ginsenoside Rg1 to mitigate DOX-induced cardiotoxicity. The Rg1-loaded sEVs exhibited a strikingly cardioprotective function compared to free Ginsenoside Rg1 and sEVs. The 3D cell culture system is a promising strategy for sEV mass production and delivery platform for future clinical research.

## Figures and Tables

**Figure 1 pharmaceutics-16-00593-f001:**
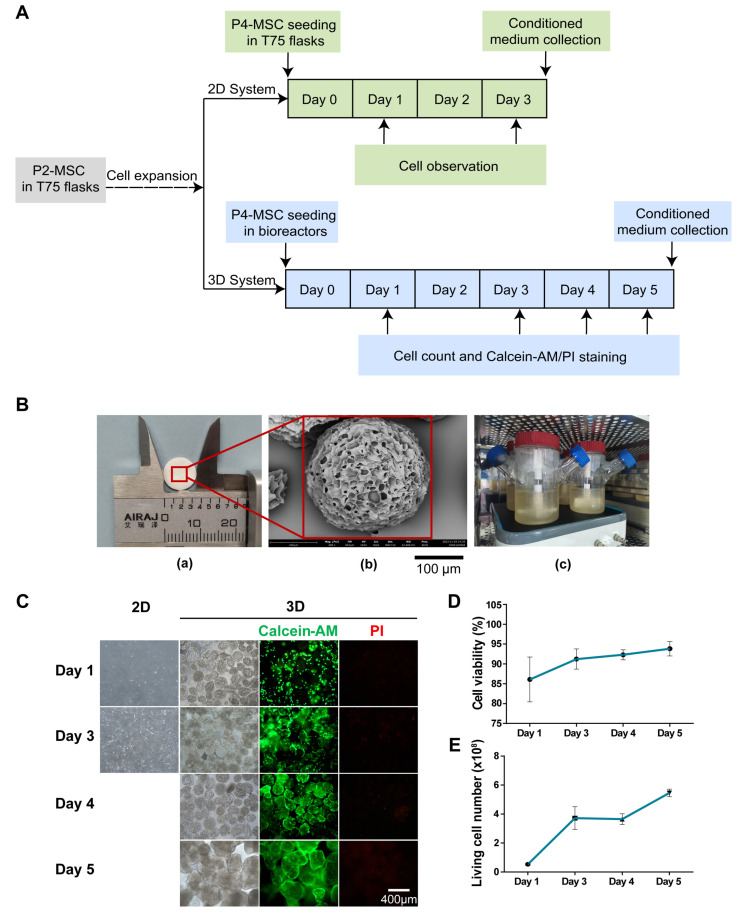
The 3D culture of MSCs. (**A**) Schematic diagram of scale-up expansion of MSCs from 2D system to 3D system. (**B**) Image of tablet of microcarriers before dissolution (**a**); SEM image of a single microcarrier. Scale bar = 100 μm. (**b**); and image of 3D spinner flasks with MSCs and microcarriers in serum-free medium (**c**). (**C**) Images of 2D-MSCs on Day 1 and Day 3 and that of 3D-MSCs on Days 1, 3, 4 and 5. The 3D-MSCs in microcarriers were visualized using Calcein-AM (green) and PI (red) staining. Scale bar = 400 μm. (**D**) Cell viability of 3D-MSCs. (**E**) The living cell number of 3D-MSCs.

**Figure 2 pharmaceutics-16-00593-f002:**
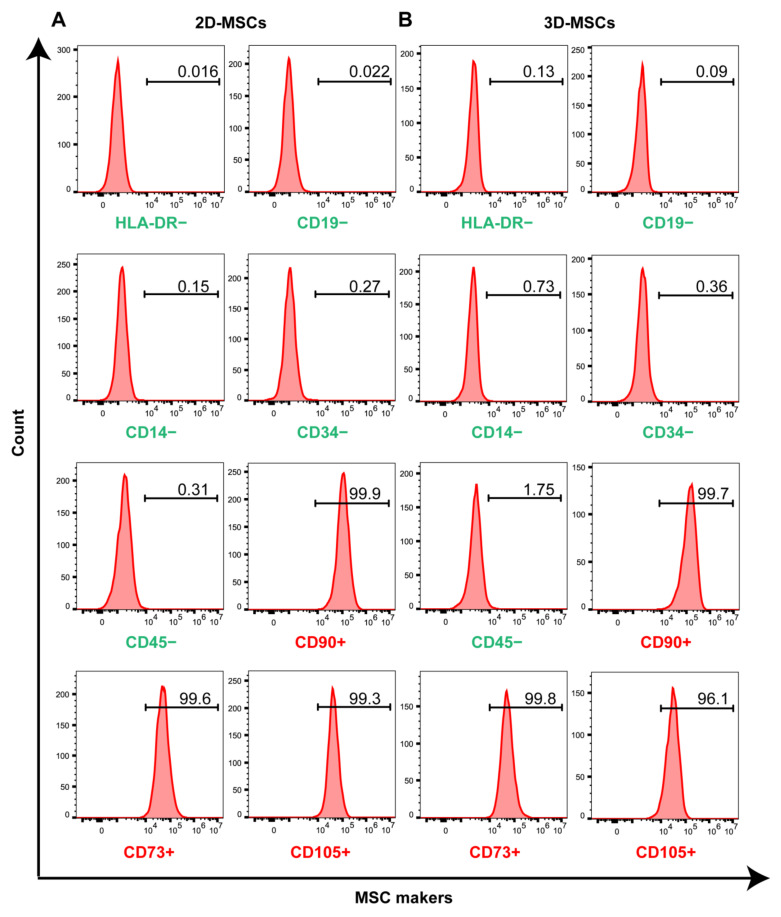
The flow cytometry analysis of 2D-MSCs (**A**) and 3D-MSCs (**B**). The negative surface markers: HLA-DR−, CD19−, CD14−, CD34− and CD45− (green). The positive surface markers: CD90+, CD73+ and CD105+ (red).

**Figure 3 pharmaceutics-16-00593-f003:**
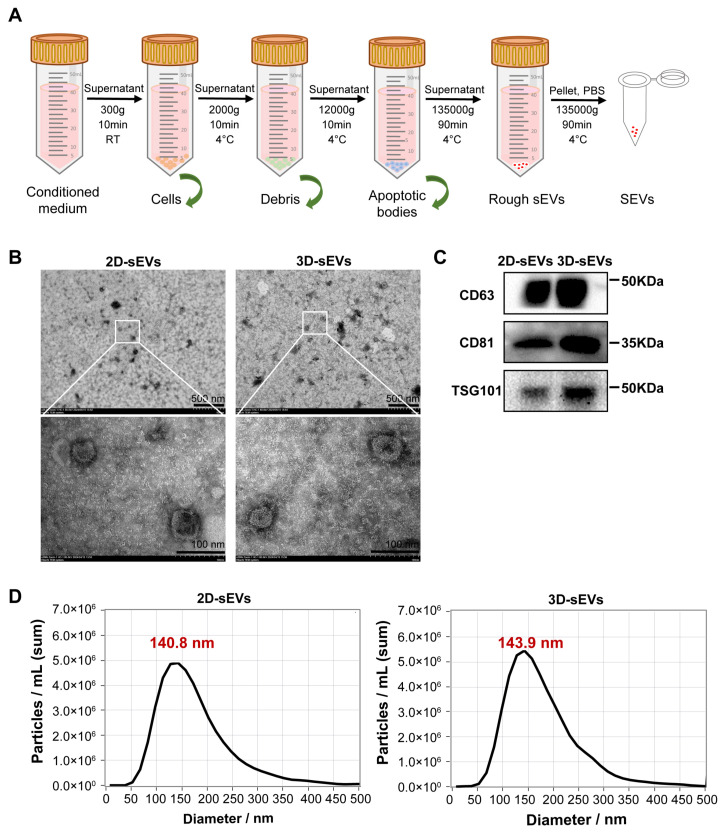
Isolation and characterization of sEVs. (**A**) Isolation of sEVs by dUC. (**B**) TEM images of 2D-sEVs and 3D-sEVs. Scale bars = 500 nm and 100 nm. (**C**) The protein expression of CD63, CD81 and TSG101 in 2D-sEVs and 3D-sEVs by WB. (**D**) The diameter distribution of 2D-sEVs (left) and 3D-sEVs (right) was analyzed by NTA.

**Figure 4 pharmaceutics-16-00593-f004:**
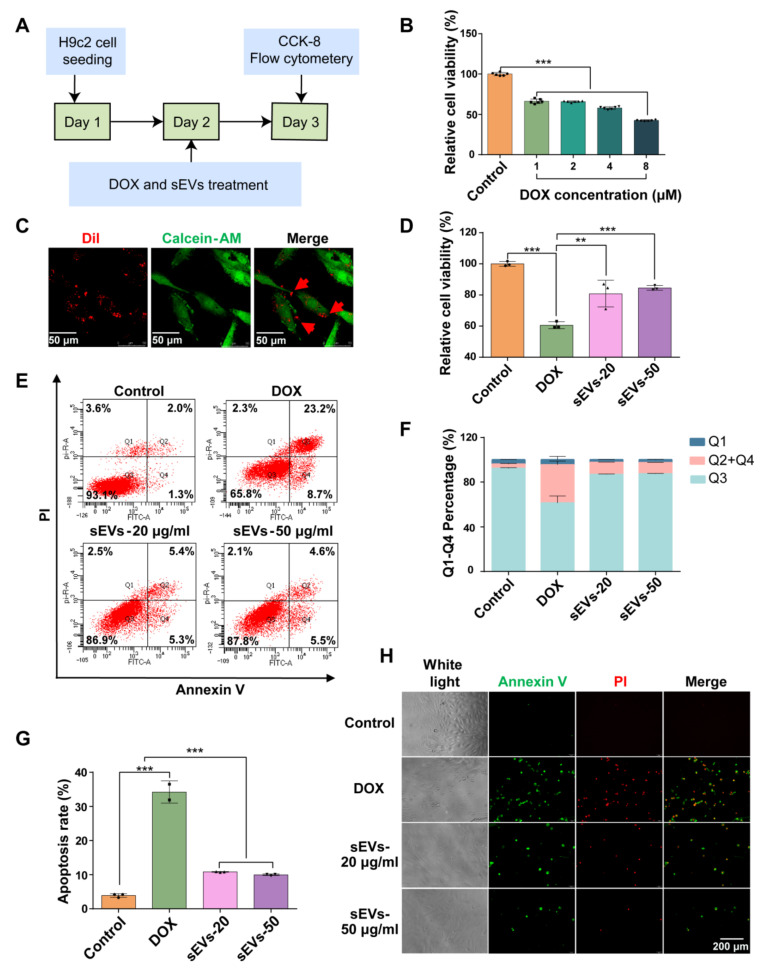
The 3D-sEVs decreased H9c2 cell apoptosis and protected H9c2 cells from DOX-induced cardio injury. (**A**) Schematic representation of the study design. (**B**) Relative cell viability of H9c2 cells treated with 1, 2, 4 and 8 μM DOX after 24 h. *n* = 6. (**C**) Dil (red)-labelled sEVs in Calcein-AM (green)-labelled H9c2 cells. Scale bar = 50 µm. (**D**) Relative cell viability of H9c2 cells after 24 h treatment. *n* = 3. (**E**) Flow cytometry detection of H9c2 cells stained with Annexin V and PI. Q1, Annexin V−/PI+, necrosis; Q2, Annexin V+/PI+, late apoptosis; Q3, Annexin V−/PI−, living cell population; Q4, Annexin V+/PI−, early apoptosis. (**F**) Quantification of the percentage of Q1, Q2 + Q4, and Q3. *n* = 3. (**G**) Quantification of the percentage of apoptotic cells (Q2 + Q4). (**H**) Annexin V (green) and PI (red) staining H9c2 cells. Scale bar = 200 µm. *** *p* < 0.001, ** *p* < 0.01.

**Figure 5 pharmaceutics-16-00593-f005:**
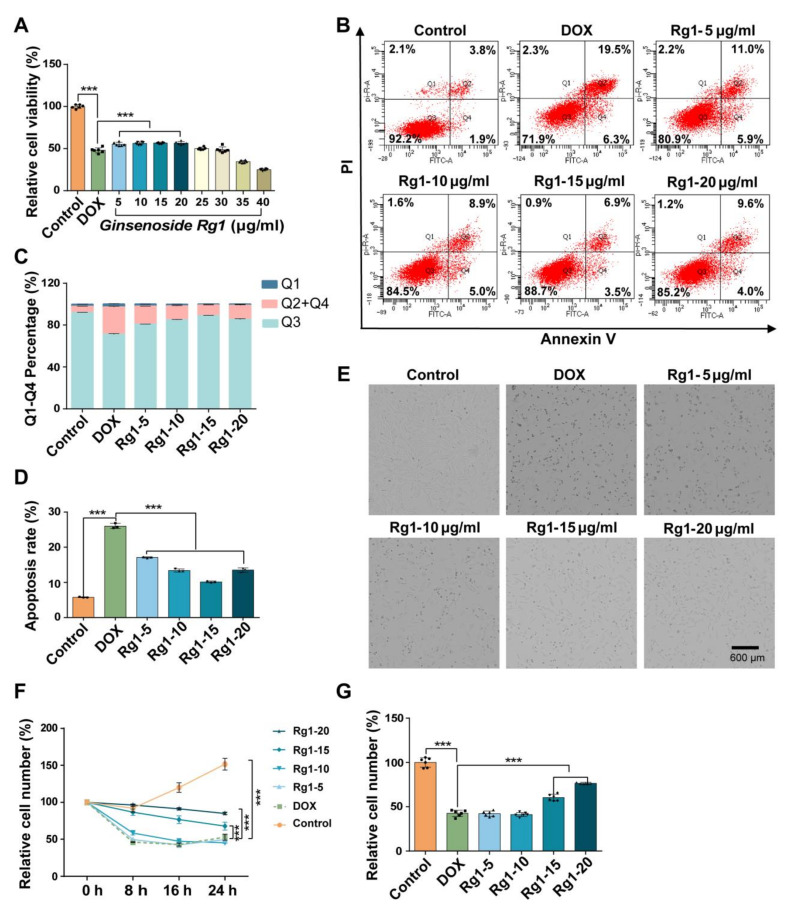
Ginsenoside Rg1 reduced DOX-induced H9c2 cell damage. (**A**) Relative cell viability of H9c2 cells with DOX and/or Rg1 for 24 h. *n* = 6. (**B**) Flow cytometry detection of Annexin V and PI stained H9c2 cells after treatments of DOX and/or Rg1. *n* = 3. (**C**) Percentage of Q1, Q2 + Q4, Q3 in Annexin V and PI stained H9c2 cells. *n* = 3. (**D**) Percentage of apoptotic cells (Q2 + Q4) in each group. *n* = 3. (**E**) Images of H9c2 cells at 24 h using ImageXpress Pico automated cell imaging system. Scale bar = 600 µm. *n* = 6. (**F**) Proliferation curve of H9c2 cells treated with DOX and/or Rg1 for 24 h. (**G**) The relative cell number of H9c2 cells at 24 h. *n* = 6. *** *p* < 0.001.

**Figure 6 pharmaceutics-16-00593-f006:**
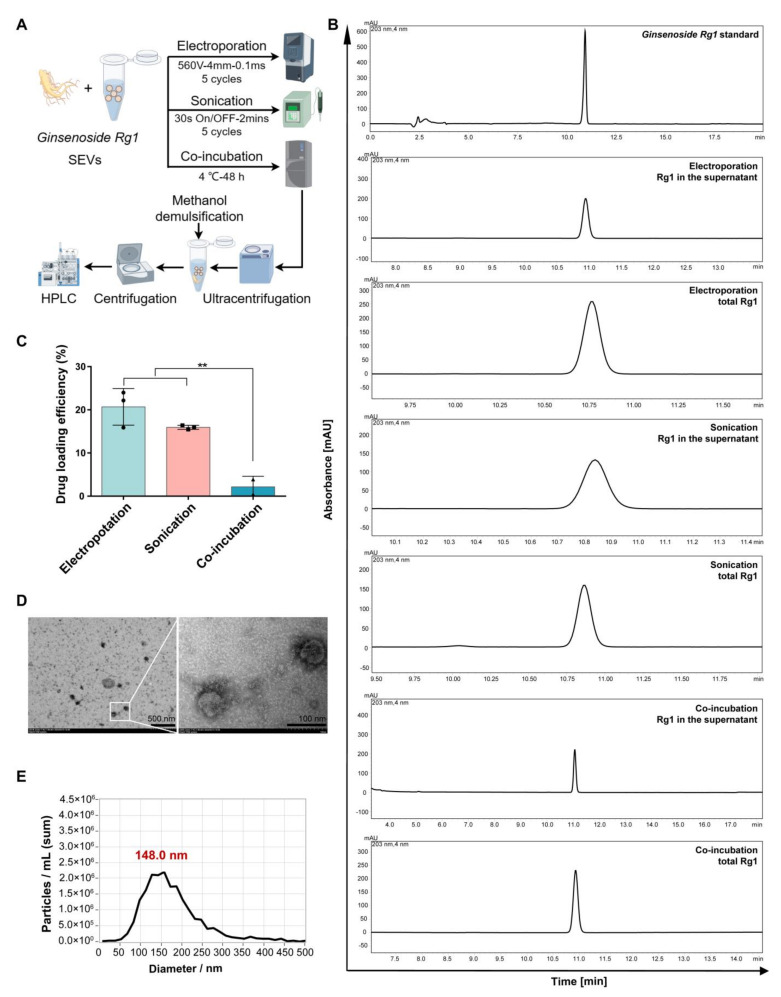
Ginsenoside Rg1 loading of 3D-sEVs. (**A**) Schematic diagram of Ginsenoside Rg1 loading with electroporation, sonication and co-incubation methods and following detection. (**B**) The HPLC chromatogram of Ginsenoside Rg1. (**C**) Drug loading efficiency of three methods. *n* = 3. (**D**) TEM images of Rg1-loaded sEVs. Scale bars = 500 nm and 100 nm. (**E**) The NTA diameter distribution of Rg1-loaded sEVs. ** *p* < 0.01.

**Figure 7 pharmaceutics-16-00593-f007:**
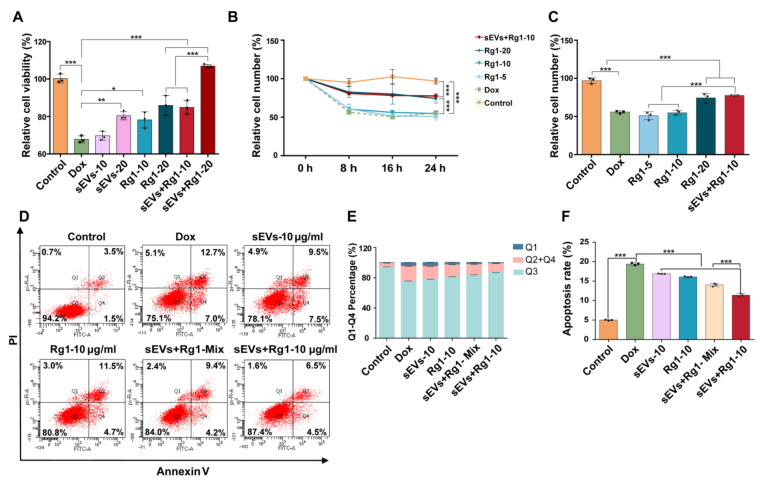
Rg1-loaded sEVs reduced DOX-induced H9c2 cell damage. (**A**) Relative cell viability of H9c2 cells treated with control, DOX, sEVs, Rg1, and sEVs + Rg1 for 24 h. *n* = 3. (**B**) Proliferation curve of H9c2 cells treated with DOX and/or Rg1 for 24 h. Cell number was pictured at each 8 h and counted using ImageXpress Pico automated cell imaging system. *n* = 3. The relative cell counts were normalized using the cell number at 0 h of each group. (**C**) The relative cell number of H9c2 cells at 24 h. (**D**) Flow cytometry detection of Annexin V and PI stained H9c2 cells after treatments of DOX and/or Rg1. *n* = 3. (**E**) Percentage of Q1, Q2 + Q4, Q3 in Annexin V and PI stained H9c2 cells. *n* = 3. (**F**) Percentage of apoptotic cells (Q2 + Q4) in each group. *n* = 3. *** *p* < 0.001, ** *p* < 0.01, * *p* < 0.05.

**Table 1 pharmaceutics-16-00593-t001:** Statistics of sEV production in 2D and 3D cultivation.

Terms	3D Culture System	2D Culture System ^#^
Cell input	8 × 10^7^ cells	
Flasks	4 spinner flasks	132 T75 flasks
Conditioned medium	2 × 10^3^ mL	1.58 × 10^3^ mL
Total volume of sEVs	3.3 × 10^3^ μL	3.96 × 10^3^ μL
Average conc. of particles	6.6 × 10^11^/mL	3.35 × 10^11^/mL
Total particle number	2.2 × 10^12^	1.33 × 10^12^
Average conc. of protein	0.53 mg/mL	0.26 mg/mL
Total protein	1.74 mg	1.04 mg
Bench time *	32 min	264 min

* The bench time included time cost on cell seeding, sampling for cell viability and number detection, and collecting conditioned medium in 2D and 3D culture at desired time points. ^#^ The data for 2D culture were calculated based on the average production of five T75 flasks.

## Data Availability

Data is contained within the article and Appendix A.

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
