# Peer review of "Mass Production of Rg1-Loaded Small Extracellular Vesicles Using a 3D Bioreactor System for Enhanced Cardioprotective Efficacy of Doxorubicin-Induced Cardiotoxicity"

_pharmaceutics, 2024, doi:10.3390/pharmaceutics16050593_

Round 1

Reviewer 1 Report

Comments and Suggestions for Authors

The research paper entitled “Mass Production of Rg1-Loaded Exosomes Using a 3D Bioreac- 2

tor System for Enhanced Cardioprotective Efficacy of DOX-In- 3 duced Cardiotoxicity” explore the possibility to upscale the prodaction of extracellular vesicles using a 3D bioreactor. It also explore the possibility to use these technology to produce extracellular vesicles from MSC  to exploit their cardiopretective effect to contrast the adverse effect of chemotherapeutic drug like Doxorubicin. To this regards I have some minor comments towards this paper:

I recommend to the authors to read carefully the last release of ISEV titled  “Minimal information for studies of extracellular vesicles (MISEV2023 PMID: 38326288): From basic to advanced approaches” in order to change the nomenclature used in the paper:” ISEV recommends use of the generic term ‘Extracellular Vescicles’ and operational extensions of this term instead of inconsistently defined and sometimes misleading terms such as ‘exosomes’ and ‘ectosomes’ that are associated with biogenesis pathways that are difficult to establish.”  

Even for the characterization of your EV follow the recommendations “Note that affinity-based protocols involving the tetraspanins CD9, CD63 and CD81 are not specific for exosomes as an EV subtype; using antibodies to each of these tetraspanins enriches EV populations that do not completely overlap in molecular composition”.

There is at least another work that highlight also the cardioprotective effect of EV loaded with DOXO please cite (PMID: 27558906).

Line 300: you write you treat H9c2 cells with DOX and exosomes “to verify the therapeutic function of exososomes”. But following reading is clear that you wanted to assess the protecting role of exosomes toward cardiotoxicity induced by DOX; moreover, H9c2 cells are not cancer cell lines so it is senseless to treat them with antitumoral drug to assess the efficacy. If it is so, correct the text according to the correct aim of the experiment. It should be interesting if you could treat a cancer cell lines (e.g. MDA-MB 231) to assess if the presence of exosomes could interfere with the DOX antitumor efficacy even in presence of Gingeroside RG1.

I also encourage the authors to accurately spellcheck the English.

Comments on the Quality of English Language

There Are some minor grammar error to fix. I suggest to accurately spellcheck before resubmit.

Reviewer 2 Report

Comments and Suggestions for Authors

there are major issues in the paper that need to be corrected. 

1) the paper does not comply with MISEV guidelines, (TEM pictures with wide and zoom, negative protein controls, please read the whole MISEV paper), this needs to be performed

2) please use correct wording (small EVs instead of exosome)

3) More generally, number of EVs and its purity seems unexpectedly high. 

- purity ratio calculated : 2.4 x 10^11 EVs/microgram => higher than any reported to my knowledge (see paper how pure are my vesicles) => is there an error in number reported ? 

- Number of EVs per cell 5.5 x 10^6 per cell in the 2D condition => seems largely higher than any reported case to my knowledge (we usually obtain about 2000-5000 EVs per cell in 2D flasks in our lab from MSCs with similar methods), it is 10^3 fold lower... 

- I suspect there is either an error in number reported, or that the vesicles are produced in a protein/EV comtaminated media. Anyway, authors should perform the same production and isolation procedure (3D flask for few days at 37 degrees, etc) but without cells, this will be a good negative control to understand whether reported numbers are real EVs or comtaminants (as i suspect as TEM pictures seems to show EVs surrounded by lot of proteins. 

minor : 

- the paper may be simplified and shorten

line 90 : Traditional Chinese medicine plays a noteworthy role in the prevention and treat-90 ment of human diseases => quite vague

line 97 : not clear : Traditional Chinese medicine plays a noteworthy role in the prevention and treat-90 ment of human diseases

line 112 : precise whether it is xeno free, contains platelet lysate, or other ? 

LINE 139 : The gold standard for exosome isolation is ultracentrifugation[16] by ultracentrifuge 139 (Beckman, USA) which requires five sequential centrifugation steps. => it is not the gold standard but the most frequent

line 223 : it is an introduction

Comments on the Quality of English Language

I am not qualified

Reviewer 3 Report

Comments and Suggestions for Authors

Reviewer’s report

MS: pharmaceutics-2931110

This study described the upscaling of mesenchymal stem cell (MSC) culture using the 3D bioreactor to improve the yields of MSC-derived extracellular vesicles (MSC-EVs). A natural compound, Rg-1-loaded MSC-EVs seemed to enhance the cardioprotective effect against doxorubicin in a mouse cardiomyocyte cell line. This study is interesting, but several checks are needed before further consideration for publication in Pharmaceuticals.

Comments

1. A broad term “extracellular vesicles (EVs)” or an operational term “small extracellular vesicles (EVs)” should be used instead of exosomes throughout the manuscript and figures. This comment is aimed at minimizing the confusion in the nomenclature of the EV studies as per MISEV2023 guidelines.

2.     The method section about the EV-drug loading efficacy (lines 174-180) was quite confusing. This section should be rewritten.

3.     Figure 1 – Cell proliferation rate or doubling time of 2D-MSC culture should be presented and compared with the 3D-MSC microcarrier culture system.

4.     How many independent biological replicates were performed for the data in Table 1? Did data for 2D culture (e.g., 132 T75 flasks) obtain from the experiments or a calculation?

5.     How many MSC-EV particle numbers are equivalent to 20 ug/mL and 50 ug/mL MSC-EVs? 

6.     Figure 4 - In addition to the blank, the control set should include either particle depletion (e.g., by ultrafiltration) or inactivation (e.g., by heating) to confirm the observed activities were from functional EVs.

7.     Fig 6 – How many replicates were performed for this experiment? The error bar should be presented.

8.     Fig 7C – please correct the y-axis. The number/scale seemed wrong.

9.     The experimental design in Fig 7 was not quite consistent. Nonetheless, Rg1-loaded MSC-EVs at 20 ug/mL (Fig 7A) seemed promising against DOX-induced cell death. This dosage should be included in the results of Figs 7B-F. Also, in all experiments, the control set should include MSC-EVs alone and Rg-1 alone as the comparators for evaluating Rg-1-loaded MSC-EVs.

10.  Limitations of the study should be added to the discussion section.

11.  There are several typos throughout the manuscript, some of which are listed here: line 78 (“20-folder” to “20-fold”), line 101 (“the field” to “the yield”), line 143 & 144 (2 “Finally” repeats), line 172 & 173 (“co-cultured” to “co-incubated”), line 222 & 258 (“3.13. D…” to “3.1. 3D…” and (“3.23. D…” to “3.2. 3D…”). Several sentences and paragraphs are difficult to understand. Extensive editing of the English language is needed.

Comments on the Quality of English Language

Extensive editing of the English language is required.

Round 2

Reviewer 2 Report

Comments and Suggestions for Authors

thanks for this revised version that correct many important mistakes

there are still few to correct : 

- we still need the TEM pictures

- this need to be added in the results and discussed :

"Response: Thanks for your kind suggestions. We have corrected the total number of sEV particles in Table 1. We have performed NTA analysis of the MSC complete medium without cells, and the MSC conditioned medium after culture in Jan. 2023. The former medium contained approximately 1.2 × 1010 /ml particles, while the latter contained approximately 2.3 × 1010 /ml particles. We believe that there was some consumption of the original vesicles in the medium and a significant production of new vesicles after MSC culture"

=>key points to discuss : 1) the production was performed in a xeno free media, in presence of platelet lysate, with a lot of platelet lysate derived EVs. 

2) it seems platelet lysate derived EVs were consumed at least partly by MSCs (state numbers)

3) meanwhile EVs from MSC were produced

4) the resulting EVs are probably a mixture of Platelet and MSC derived EVs, and the exact proportion of both is not known. 

=> change conclusion/discussion accordingly, or provide data to show that EVs are mostly MSC derived

best regards

Reviewer 3 Report

Comments and Suggestions for Authors

The authors fully addressed all my comments appropriately. I have no further questions or concerns.

Comments on the Quality of English Language

Moderate editing of English language required

Round 3

Reviewer 2 Report

Comments and Suggestions for Authors

ok